# Time-reversal symmetry, anomalies, and dualities in (2+1)$d$

Clay Córdova[1], Po-Shen Hsin[2] and Nathan Seiberg[1][⋆]

**1** School of Natural Sciences, Institute for Advanced Study, Princeton, NJ, USA
**2** Physics Department, Princeton University, Princeton, NJ, USA

⋆ seiberg@ias.edu

## Abstract

We study continuum quantum field theories in 2+1 dimensions with time-reversal symmetry $\mathcal{T}$. The standard relation $\mathcal{T}^2 = (-1)^F$ is satisfied on all the "perturbative operators" i.e. polynomials in the fundamental fields and their derivatives. However, we find that it is often the case that acting on more complicated operators $\mathcal{T}^2 = (-1)^F \mathcal{M}$ with $\mathcal{M}$ a non-trivial global symmetry. For example, acting on monopole operators, $\mathcal{M}$ could be $\pm 1$ depending on the magnetic charge. We study in detail $U(1)$ gauge theories with fermions of various charges. Such a modification of the time-reversal algebra happens when the number of odd charge fermions is $2 \bmod 4$, e.g. in QED with two fermions. Our work also clarifies the dynamics of QED with fermions of higher charges. In particular, we argue that the long-distance behavior of QED with a single fermion of charge 2 is a free theory consisting of a Dirac fermion and a decoupled topological quantum field theory. The extension to an arbitrary even charge is straightforward. The generalization of these abelian theories to $SO(N)$ gauge theories with fermions in the vector or in two-index tensor representations leads to new results and new consistency conditions on previously suggested scenarios for the dynamics of these theories. Among these new results is a surprising non-abelian symmetry involving time-reversal.



# 1  Introduction

Time-reversal symmetry is an important property of a variety of systems relevant to both high-energy and condensed matter physics. In this paper we clarify some aspects of time-reversal symmetry in gauge theories. Our methodology here is that most natural in continuum field theory. We will start with a continuum Lagrangian defining our model at short distances and try to ascertain its long-distance behavior. A first step in this analysis is a precise determination of the global symmetry of the model. This includes the ordinary unitary global symmetries, as well as spacetime symmetries such a time-reversal. As we describe below, in general these symmetries are linked in a non-trivial algebra.

    We will focus here on three-dimensional systems based on some gauge group and fermions transforming in some representation. We will mostly study $U(1)$ and $SO(N)$ gauge theories ($U(1)$ is a special case of $SO(N)$ with $N = 2$, but with many special features) and fermions in the vector or the two index tensor of $SO(N)$ (in $U(1)$ these are fermions of charge 1 or 2). In order to determine the IR behavior of the system we need to understand in detail its symmetries and in particular its time-reversal symmetry.

## 1.1  $\mathcal{T}$

Time-reversal symmetry $\mathcal{T}$ is an antiunitary transformation that acts on the time coordinate as $t \rightarrow -t$ combined with some action on the fields in the theory. In Euclidean spacetime it reverses the orientation of spacetime. In general, the $\mathcal{T}$ symmetry of the theory is not unique. We can redefine $\mathcal{T}$ by combining it with a global symmetry transformation. For example, many systems have a unitary symmetry that acts as an outer automorphism of the gauge group, which is called charge conjugation $\mathcal{C}$. Then we can say that the basic time-reversal symmetry is $\mathcal{T}$ or $\mathcal{C}\mathcal{T}$.

    Neither of these choices is universally natural. For instance, the standard definition of time-reversal in four-dimensional free Maxwell theory acts on the electric and magnetic fields as $E \rightarrow E$ and $B \rightarrow -B$, while charge conjugation reverses the sign of both. However, electromagnetic duality exchanges $E$ and $B$ and therefore maps $\mathcal{T}$ to $\mathcal{C}\mathcal{T}$. In this paper we will follow [1, 2] and define the symmetry $\mathcal{T}$ to act on a gauge field $a$ as $\mathcal{T}(a(t)) = a(-t)$. In components this reads

$$\mathcal{T}(a_0(t)) = -a_0(-t) \,, \qquad \mathcal{T}(a_i(t)) = a_i(-t) \,. \tag{1.1}$$

One advantage of the above is that it makes sense even for systems where there is no natural notion of charge conjugation. (Note that if the $U(1)$ gauge field is that of ordinary electromagnetism, this symmetry is usually called $\mathcal{C}\mathcal{T}$.) In the condensed matter literature on models

with a global $U(1)$ our convention (1.1) is known as $U(1) \times \mathbb{Z}_2^{\mathcal{T}}$ as opposed to $U(1) \rtimes \mathbb{Z}_2^{\mathcal{T}}$ (see e.g. [2–5]). Notice that as a consequence of these definitions, the electric charge $Q$ is odd under $\mathcal{T}$, while the magnetic charge $M$ is even.

$$\mathcal{T} Q \mathcal{T}^{-1} = -Q \,, \qquad \mathcal{T} M \mathcal{T}^{-1} = M \,. \qquad (1.2)$$

In particular, this means that on a charged fermion field $\psi$ in an abelian gauge theory, we have:

$$\mathcal{T}(\psi) = \gamma_0 \psi(-t)^* \,, \qquad \mathcal{CT}(\psi) = \gamma_0 \psi(-t) \,. \qquad (1.3)$$

Although time-reversal is an antiunitary symmetry, the operator $\mathcal{T}^2$ is a unitary symmetry. In systems that depend on spin structure (like the models with fermions of interest here) there is also a fermion number symmetry $(-1)^F$, and this leads to several elementary possibilities for the unitary symmetry $\mathcal{T}^2$.

- Non-spin theories with $\mathcal{T}^2 = 1$. We refer to this as $\mathbb{Z}_2^{\mathcal{T}}$. In Euclidean signature this symmetry algebra means that the theory may be formulated on any unorientable manifold. These systems can have an 't Hooft anomaly for the time-reversal symmetry valued in $\mathbb{Z}_2 \times \mathbb{Z}_2$ [6–9].

- Spin theories with $\mathcal{T}^2 = (-1)^F$. We refer to this as $\mathbb{Z}_4^{\mathcal{T}}$. This is also the algebra realized on the charged fermions in (1.3). In Euclidean signature this symmetry algebra means that the theory may be formulated on unorientable manifolds with a $Pin^+$ structure. Any system with this symmetry has an 't Hooft anomaly $\nu \in \mathbb{Z}_{16}$ characterizing its behavior on such manifolds [1, 3, 4, 10, 11].

- Spin theories with $\mathcal{T}^2 = 1$. We refer to this as $\mathbb{Z}_2^{\mathcal{T}} \times \mathbb{Z}_2^F$. In Euclidean signature this symmetry algebra means that the theory may be formulated on unorientable manifolds with a $Pin^-$ structure. Unlike the cases above, there are no possible 't Hooft anomalies for this symmetry algebra [12].

As we describe below, the possibilities listed above are by no means exhaustive, and we give examples of time-reversal invariant gauge theories where $\mathcal{T}^2$ is a more general unitary symmetry. Similar phenomena have been observed in [4, 10].

One particularly interesting class of time-reversal invariant theories are certain spin topological field theories defined by Chern-Simons gauge theories at specific non-zero values of the level. These models are not classically time-reversal invariant but they enjoy level-rank duality that changes the sign of the level and hence defines a $\mathcal{T}$ symmetry of quantum theory satisfying $\mathcal{T}^2 = (-1)^F$ [13–15].[1] A summary of these theories and their associated value of $\nu$ is given in table 1.

## 1.2 What is the Global Symmetry?

As discussed above, the models of interest to us in this paper all admit an ultraviolet definition as a gauge theory with gauge group $H$. To analyze their global symmetry group $G$, it is often useful to discuss the related model defined by restricting the dynamical gauge fields to be classical.

In this theory we have a set of fields with a global symmetry $K$. The global symmetry action is characterized by some 't Hooft anomaly. This means that in the presence of background $K$ gauge fields, the system is not gauge invariant and this lack of gauge invariance cannot be

---

[1]Certain special cases of the level also define bosonic TQFTs. However, in general the dualities below only hold when the theories are promoted to spin theories [13, 14].

| $\mathcal{T}$-invariant spin-TQFT | Anomaly $\nu$ (mod 16) |
|:---:|:---:|
| $U(n)_{n,2n}$ | 2 |
| $Sp(n)_n$ | $2n$ |
| $SO(n)_n$ | $n$ |
| $O(n)^1_{n,n+3}$ | $n$ |
| $O(n)^1_{n,n-1}$ | $n$ |

Table 1: Time-reversal invariant spin TQFTs and their associated anomaly $\nu$. These anomalies have been computed by various methods [8, 16–20]. The notation for the levels in the $O(n)$ Chern-Simons theory is explained in [15]. The anomaly for the $O(n)^1$ theories is determined based on the consistency of the conjectured phase diagrams of [15, 20]. Note that in general redefining the orientation of spacetime changes $\nu \to -\nu$ [1]. For a given theory the sign is convention dependent, but the relative sign between theories is meaningful. For instance, in the $U(n)_{n,2n}$ sequence it is natural to fix the sign to be $(-1)^{n-1}$ [20]. Some of the TQFTs above also admit unitary global symmetries of order two and these can be combined with $\mathcal{T}$ to produce other antiunitary symmetries of the model with a different value of $\nu$. An example that will occur below is the T-Pfaffian theory vs. the CT-Pfaffian theory [3–5]. In addition, the value of $\nu$ can depend on other choices like the eigenvalue of $\mathcal{T}^2$ on the anyons. Several special cases of these theories have been previously considered. The example $SO(2)_2 \leftrightarrow U(1)_{1,2}$ is known as the semion-fermion [3]. Meanwhile $SO(3)_3$ was analyzed in [3], and $SO(4)_4$ was studied in [18, 19, 21]. The theory $O(2)_{2,1}$ is equivalent to the T-Pfaffian theory [15, 22, 23], and we have the duality $O(2)_{2,5} \leftrightarrow U(2)_{2,4}$ [15].

fixed by adjusting any local term. Instead, the anomaly is characterized by a local term in one higher dimension.

Next, we try to gauge a subgroup $H \subset K$. This can be done only when the anomaly vanishes when restricted to $H$ gauge fields. What is the global symmetry after this gauging? In many cases it is given by the group

$$G \cong N(H,K)/H \,, \tag{1.4}$$

where $N(H,K)$ is the normalizer of $H$ in $K$, and we mod out by the gauged subgroup $H$.

As an example of this construction that will occur below, we can start with $2N_f$ real Majorana fermions and then $K \cong O(2N_f)$. The subgroup $H \cong U(1)$ that acts on the fields as Dirac fermions of unit charge is anomaly free and can be gauged. The resulting global symmetry group $G$ is then $PSU(N_f) \rtimes \mathbb{Z}_2^{\mathcal{C}}$, where $\mathcal{C}$ is a charge conjugation symmetry.

There are two phenomena that can make the answer (1.4) wrong:

- The anomaly might mean that if the gauge fields of $H$ are dynamical, then the some of the elements in $N(H,K)$ are no longer a symmetry. This means that only a subgroup of $G$ is the true global symmetry.

- When the gauge fields of $H$ are dynamical, we can have a new emergent symmetry $\widehat{H}$. Examples that we will see below are that in 2+1d when $H \cong U(1)$ we have an emergent magnetic symmetry $\widehat{H} \cong U(1)$. Similarly, when $H \cong SO(N)$ we have an emergent magnetic symmetry $\widehat{H} \cong \mathbb{Z}_2$.

These two phenomena often mix with each other. One aspect of this is that the symmetries $\widehat{H}$ and $G$ can form a nontrivial algebra. We refer to this possibility by saying that $G$ is *deformed* by $\widehat{H}$ (examples were studied in [15, 21, 24]). For instance, we will describe systems with a time-reversal symmetry $\mathcal{T} \in G$ where the unitary symmetry $\mathcal{T}^2$ is neither of the two elementary

possibilities discussed above (i.e. $\mathcal{T}^2 = (-1)^F$ or $\mathcal{T}^2 = 1$), but instead is an element of the emergent magnetic symmetry $\mathcal{T}^2 \in \widehat{H}$. These symmetry algebras have been explored in [4,10]. We will also see examples where the time-reversal symmetry $\mathcal{T}$ participates in a non-abelian algebra with elements of $\widehat{H}$ and $G$.

### 1.3 Monopole Operators and Their Quantum Numbers

Many of our results follow from a careful analysis of monopole operators in abelian gauge theory and their quantum numbers under various global symmetries.

Consider a $U(1)$ gauge theory with gauge field $b$ coupled to Dirac fermions $\psi^i$ of charge $q_i$. We follow standard conventions and label theories by an effective level $k$ defined for massless fermions. The effective level is partitioned into two parts. The first is the integral bare level $k_{bare} \in \mathbb{Z}$, which controls the level in the UV Lagrangian. The second piece is in general half-integral and encodes the contribution from the fermions. The level shifts under mass deformation as shown below.

$$
\begin{array}{ccc}
m_\psi < 0 & m_\psi = 0 & m_\psi > 0 \\[2mm]
k_{bare} & k \equiv k_{bare} + \frac{1}{2}\sum_i q_i^2 & k_{bare} + \sum_i q_i^2
\end{array}
\tag{1.5}
$$

Note that when the fermions are massive, the level is always an integer.

In addition to the dynamical gauge field $b$, it is also instructive to introduce a background gauge field $A$, which couples to all fermions with charge one. As in [25], $A$ can be viewed as a $spin^c$ connection, and we can include a mixed Chern-Simons term for $A$ and $b$ in the theory with bare level $Q$. Thus Lagrangian of interest is

$$
\mathcal{L} = \frac{Q}{2\pi} b dA + \frac{k_{bare}}{4\pi} b db + i\overline{\psi}^i (\slashed{\partial} + \slashed{A} + q_i \slashed{b})\psi^i \ .
\tag{1.6}
$$

Our focus is on time-reversal invariant theories, which must have vanishing effective levels and hence we adjust the counterterms to

$$
Q = -\frac{1}{2}\sum_i q_i \ , \qquad k_{bare} = -\frac{1}{2}\sum_i q_i^2 \ .
\tag{1.7}
$$

Since $Q$ and $k_{bare}$ must be integers this means that $\mathcal{T}$ symmetry requires that the number $N_{odd}$ of fermions with odd charge $q_i$ must be even [26–28].

The fact that all the elementary fermions carry charge one under the background field $A$ and all the elementary bosons are neutral means that all of these models superficially satisfy the spin/charge relation stating that all the fermions carry odd charge under $A$ and all the bosons have even charge. More mathematically, this is the statement that if $A$ is a $spin^c$ connection, we can formulate the theory without a choice of a spin structure (or even on a non-spin manifold), but with a choice of a $spin^c$ structure. However, although this is true for all the perturbative states, we should also examine monopole operators. The condition that the spin/charge relation is satisfied for them is [25]

$$
Q = k_{bare} \quad \mathrm{mod}\ 2 \ .
\tag{1.8}
$$

Now let us turn to the action of time-reversal. On operators constructed from the elementary fields, we have the standard relation $\mathcal{T}^2 = (-1)^F$. However on states carrying magnetic charge this relation can be modified. In [2,4,10] our dynamical gauge field $b$ was interpreted as a classical background field and a mixed anomaly was found between time-reversal symmetry $\mathcal{T}$ and the $U(1)$ global symmetry coupling to $b$. When the field $b$ is instead dynamical

we interpret this result to mean that the symmetry algebra is modified as in the discussion of section 1.2. Specifically we find:

$$\mathcal{T}^2 = \begin{cases} (-1)^F \mathcal{M} & N_{odd} = 2 \bmod 4 \, , \\ (-1)^F & N_{odd} = 0 \bmod 4 \, , \end{cases} \tag{1.9}$$

where in the above,

$$\mathcal{M} \equiv (-1)^M \tag{1.10}$$

generates the $\mathbb{Z}_2$ subgroup of the $U(1)$ magnetic global symmetry. Notice also that $N_{odd}/2 = k_{bare} \bmod 2$, and thus the above relation can also be expressed by saying that $\mathcal{T}^2$ contains the magnetic symmetry whenever $k_{bare}$ is odd. We review aspects of this result as they arise below.

### 1.4 Summary of Models

Our first class of examples is three-dimensional quantum electrodynamics $U(1)_0$ with $N_f$ fermionic flavors of unit charge, where as described above we choose $N_f$ to be even to ensure time-reversal symmetry.

The global symmetries of these models are easily diagnosed. The continuous part is $(SU(N_f) \times U(1)_M)/\mathbb{Z}_{N_f}$, where the $U(1)_M$ factor is the magnetic global symmetry that acts on monopole operators. Additionally we have charge conjugation $\mathcal{C}$ and time-reversal $\mathcal{T}$.

Following the discussion around (1.9), the unitary symmetry $\mathcal{T}^2$ depends on the number of flavors. Specifically we find that

$$N_f = 0 \bmod 4 \implies \mathcal{T}^2 = (-1)^F \, , \qquad N_f = 2 \bmod 4 \implies \mathcal{T}^2 = (-1)^F \mathcal{M} \, . \tag{1.11}$$

In the latter case time-reversal is an order four symmetry (denoted $\mathbb{Z}_4^{\mathcal{T}}$) and is mixed with the magnetic symmetry as $(U(1)_M \rtimes \mathbb{Z}_4^{\mathcal{T}})/\mathbb{Z}_2$. We also observe that, although there are fermions in the ultraviolet Lagrangian, these models do not have any gauge invariant local fermionic operators.

The special case $N_f = 2$ is worthy of separate analysis. For this theory there is a conjectured self-duality [13, 29] which implies that the IR limit has enhanced global symmetry [13, 21, 29, 30]. Including both $\mathcal{C}$ and $\mathcal{T}$ the pattern of enhancement is

$$\text{UV} : \frac{SU(2) \times Pin^-(2) \rtimes \mathbb{Z}_4^{\mathcal{T}}}{\mathbb{Z}_2 \times \mathbb{Z}_2} \longrightarrow \text{IR} : \frac{O(4) \rtimes \mathbb{Z}_4^{\mathcal{T}}}{\mathbb{Z}_2} \, . \tag{1.12}$$

Some aspects of these models have been investigated in [2] and in related analysis in the condensed matter literature [4, 10]. Note that compared to these works we do not say that time-reversal symmetry is anomalous. Indeed in all of these theories, $\mathcal{T}$ is a global symmetry of the model and the spectrum is organized into associated representations. However, if $N_f = 2 \bmod 4$, the symmetry $\mathcal{T}$ satisfies the non-standard algebra stated in (1.11). Thus in this case it is not meaningful to compute the quantity $\nu$. Instead we must separately classify and compute anomalies for the correct symmetry $(U(1)_M \rtimes \mathbb{Z}_4^{\mathcal{T}})/\mathbb{Z}_2$.

Our next class of time-reversal invariant gauge theories is QED with a single fermion of even charge $q$. These theories have a $U(1)_M$ magnetic symmetry as well as charge conjugation $\mathcal{C}$ and time-reversal $\mathcal{T}$, which obey a familiar symmetry algebra. In particular:

$$\mathcal{T}^2 = (-1)^F \, , \qquad \mathcal{C}^2 = 1 \, , \qquad \mathcal{T}\mathcal{C}\mathcal{T}^{-1} = \mathcal{C} \, , \qquad \mathcal{T} \exp(i\alpha M)\mathcal{T}^{-1} = \exp(-i\alpha M) \, . \tag{1.13}$$

Taking as input the dualities in [5, 31–33] we derive new fermionic particle-vortex dualities which determine the long-distance behavior of these models. For instance, in the simplest

case of a charge two fermion, the infrared is a free Dirac fermion together with a decoupled topological field theory $U(1)_2$.

$$U(1)_0 + \psi \text{ with charge two} \quad \longleftrightarrow \quad \text{free Dirac fermion } \chi + U(1)_2 \,, \qquad (1.14)$$

where the time-reversal symmetry in the UV acts on the topological sector in the IR via level-rank duality as in table 1.

We deform the theory (1.14) by adding monopole operators to the Lagrangian, which breaks the magnetic symmetry to $\mathbb{Z}_2$ generated by $(-1)^M = \mathcal{M}$ and preserves a new antiunitary time-reversal symmetry $\mathcal{T}'$. The algebra of these symmetries is non-abelian. In particular:

$$\mathcal{T}'\mathcal{C}\mathcal{T}'^{-1} = \mathcal{C}\mathcal{M} \,, \qquad \mathcal{T}'^2 = (-1)^F \,, \qquad (\mathcal{C}\mathcal{T}')^2 = (-1)^F \mathcal{M} \,. \qquad (1.15)$$

Both the models described above admit extensions to higher rank gauge theories. One possible such extension is to consider $U(N)$ gauge theory. Instead here we will examine $SO(N)$. In this case for $N > 2$ the magnetic symmetry is $\mathbb{Z}_2$ with generator $\mathcal{M}$.

In the case of $\text{QED}_3$ with $N_f$ flavors the natural generalization is to $SO(N)$ Chern-Simons theory coupled to $N_f$ fermions in the vector representation. These theories have been recently studied in [14, 15, 34] and participate in many dualities. These models have an $O(N_f)$ flavor symmetry with flavor charge conjugation symmetry $\mathcal{C}_f$ as well as the $\mathbb{Z}_2$ magnetic symmetry $\mathcal{M}$, and we demonstrate that their algebra with time-reversal is non-abelian

$$\mathcal{T}\mathcal{C}_f\mathcal{T}^{-1} = \mathcal{C}_f\mathcal{M} \,. \qquad (1.16)$$

Analogously, $\text{QED}_3$ with a charge two fermion naturally generalizes to $SO(N)_0$ coupled to a fermion in the symmetric tensor representation. These models have global symmetries $\mathcal{C}$, $\mathcal{M}$, and $\mathcal{T}$ and we demonstrate that the algebra is non-abelian

$$\mathcal{T}\mathcal{C}\mathcal{T}^{-1} = \mathcal{C}\mathcal{M} \,. \qquad (1.17)$$

As we describe below, this algebra is intimately related to the jump across tensor transitions of certain discrete $\theta$-parameters in these models [15]. We study the algebra (1.17) in the context of the phase diagram of these theories determined in [34], and compute the time-reversal anomaly $\nu$ using both the UV and IR descriptions.

We also briefly discuss similar theories of $SO(N)_0$ coupled to adjoint fermions, which also enjoy the algebra (1.17).

The outline of this paper is as follows. In section 2 we analyze $\text{QED}_3$ with $N_f$ fermions of unit charge and derive the algebra (1.9) by analyzing the monopole operators. In section 3, we consider $\text{QED}_3$ with a single fermion of general even charge, and we determine its long-distance behavior both with and without monopole operator deformations. In section 4 we consider $SO(N)_0$ coupled to $N_f$ vector fermions and derive the non-abelian algebra (1.16). Finally, in section 5 we analyze $SO(N)_0$ theories with tensor fermions. We demonstrate the algebra (1.17), and elucidate its interplay with the IR phase diagram.

## 2 QED$_3$ with $N_f$ Fermions of Charge One

In this section we study $U(1)$ gauge theories with time-reversal symmetry. We consider models with $N_f$ species of fermions $\psi^i$ of charge one, where $i$ is a flavor index. We will be interested in the global symmetry group, including the interplay of unitary symmetries and time-reversal. As reviewed in the introduction, we must have $N_f \in 2\mathbb{Z}$ to have $\mathcal{T}$ symmetry.

The unitary global symmetries of this model have been analyzed in detail in [21]. They form the group

$$\frac{SU(N_f) \times U(1)_M}{\mathbb{Z}_{N_f}} \rtimes \mathbb{Z}_2^{\mathcal{C}} \,, \tag{2.1}$$

where the $SU(N_f)$ subgroup acts on the fundamental fields, the $U(1)_M$ factor is the monopole symmetry, and $\mathbb{Z}_2^{\mathcal{C}}$ is the charge conjugation symmetry. Finally the $\mathbb{Z}_{N_f}$ subgroup defining the quotient is generated by:

$$(e^{2\pi i/N_f} \mathcal{I}_{N_f}, -1) \in SU(N_f) \times U(1)_M \,, \tag{2.2}$$

where $\mathcal{I}_{N_f}$ is the identity matrix. Note that this quotient does not lead to the group $U(N_f)$.

Let us review the derivation of (2.1). On the elementary fields the magnetic symmetry does not act and the $PSU(N_f) \rtimes \mathbb{Z}_2^{\mathcal{C}}$ symmetry is given by the construction described around (1.4). Meanwhile, the precise global form of the group including the magnetic symmetry $U(1)_M$ can be determined by a careful analysis of the monopole operators. It is instructive to first view the gauge field as classical, and to work out the spectrum including charged operators. We then restore the fact that the gauge field is dynamical, and select the gauge invariant local operator.

In the background of a minimally charged monopole, each fermion $\psi^i$ and $\overline{\psi}_i$ has a single zero mode with spatial wavefunction $\rho(x)$. Considering only the zero modes, we expand the fields as

$$\psi^i = \alpha^i \rho(x) \,, \qquad \overline{\psi}_i = \alpha_i^\dagger \gamma_0 (\rho(x))^* \,, \tag{2.3}$$

where $\alpha^i$ and $\alpha_i^\dagger$ are creation and annihilation operators that have equal time commutation relations $\{\alpha^i, \alpha_j^\dagger\} = \delta_j^i$. Since the fields have charge $\pm 1$, these creation and annihilation operators have no spin.

We now quantize this spectrum of zero modes. Let $|0\rangle$ denote the bare monopole state defined to be annihilated by $\alpha_i^\dagger$ for all $i$. It has zero spin, and electric charge $k_{bare} = -N_f/2$. Quantizing the Clifford algebra of zero modes leads to the state space

$$|0\rangle \,, \qquad \alpha^{i_1}|0\rangle \,, \qquad \alpha^{i_1}\alpha^{i_2}|0\rangle \,, \qquad \cdots \qquad \alpha^{i_1}\alpha^{i_2}\cdots\alpha^{i_{N_f}}|0\rangle \,. \tag{2.4}$$

We then define monopole operators $\mathfrak{M}^{i_1 i_2 \cdots i_\ell}$ via the associated state

$$|\mathfrak{M}^{i_1 i_2 \cdots i_\ell}\rangle \equiv \alpha^{i_1}\alpha^{i_2}\cdots\alpha^{i_\ell}|0\rangle \,. \tag{2.5}$$

These operators transform in totally antisymmetric representations of $SU(N_f)$ with $\ell$ indices, and they are all bosonic. The gauge invariant monopole operator has $\ell = N_f/2$. Note that under the center of $SU(N_f)$, this has $N_f$-ality $N_f/2$, which leads to the quotient (2.2).

Having understood the monopole operators, we continue our investigation of the symmetries. Observe that gauge invariant operators constructed out of the elementary fermions or gauge field strengths must have an even number of fermions and are therefore bosons. As our analysis of the monopole operators illustrates they are also bosons. Therefore we conclude that all gauge invariant local operators in this theory are bosons. Another way to see this is that the theory satisfies the spin/charge relation where the dynamical gauge field is a $spin^c$ connection, and thus gauge invariant local operators must have integral spin [25].

We now turn to our main focus which is the time-reversal symmetry $\mathcal{T}$ of these models. On the elementary gauge non-invariant fermion fields this symmetry acts as stated in (1.3). This shows that on operators built from fundamental fermions we have a standard algebra $\mathcal{T}^2 = (-1)^F$.

To determine the action of $\mathcal{T}$ in sectors with non-vanishing monopole number we observe that the sum of the operators $\mathfrak{M}^{i_1 i_2 \cdots i_\ell}$ for general $\ell$ form a Clifford algebra representation

for $Spin(2N_f)$, and the operator $\mathcal{T}$ is an anti-linear involution on this spinor representation. The operator $\mathcal{T}^2$ is then either $+1$ or $-1$ depending on whether the representation is real or pseudoreal. Combining this with the discussion of the spin above we deduce

$$\mathcal{T}^2 = \begin{cases} (-1)^F \mathcal{M} & N_f = 2 \bmod 4 \,, \\ (-1)^F & N_f = 0 \bmod 4 \,, \end{cases} \tag{2.6}$$

where $\mathcal{M} = (-1)^M$.

More explicitly, time-reversal symmetry organizes the spectrum of fermion zero-modes into singlets and Kramers doublets. Using the mode expansion (2.3) we see that the creation and annihilation operators are related as $\mathcal{T}\alpha^i\mathcal{T}^{-1} = \alpha_i^\dagger$. From this it follows that the bare monopole $|0\rangle$ is mapped by $\mathcal{T}$ to the top state in (2.4). More generally, time-reversal acts on the monopole operators as

$$\mathcal{T}\mathfrak{M}^{i_1 i_2 \cdots i_\ell}\mathcal{T}^{-1} = \frac{(-1)^{\frac{\ell(\ell-1)}{2}}}{(N_f - \ell)!}\varepsilon_{i_1 i_2 \cdots i_\ell \ j_1 j_2 \cdots j_{N_f - \ell}}\mathfrak{M}^{j_1 j_2 \cdots j_{N_f - \ell}} \,. \tag{2.7}$$

Notice that time-reversal changes fundamental indices of $SU(N_f)$ into antifundamental indices. The gauge invariant monopole operator is in a representation of $SU(N_f)$ that is isomorphic to its complex conjugate and therefore (2.7) maps the monopole operator to itself. Using this formula it is straightforward to reproduce (2.6).

It is also instructive to consider a deformation of this theory that reduces the global symmetry. We add to the Lagrangian an operator of monopole number two:

$$\delta\mathcal{L} = \delta_{i_1 j_1}\delta_{i_2 j_2}\cdots\delta_{i_{N_f/2} j_{N_f/2}}\mathfrak{M}^{i_1 i_2 \cdots i_{N_f/2}}\mathfrak{M}^{j_1 j_2 \cdots j_{N_f/2}} + c.c. \,. \tag{2.8}$$

This deformation breaks the $U(1)_M$ symmetry down to a $\mathbb{Z}_2$ subgroup generated by $\mathcal{M}$. It also breaks the flavor symmetry (2.1) down to the subgroup that preserves $\delta_{ij}$, which is the orthogonal group $O(N_f)$. In particular, it preserves a $\mathbb{Z}_2$ charge conjugation element $\mathcal{C}_f$ that acts by reflection on one of the flavor indices. Of course, there are other ways to contract the indices in the double monopole (2.8). For example, for $N_f = 0 \bmod 4$ we can replace all $\delta_{ij} \to J_{ij}$ with the standard antisymmetric $J_{ij}$ to break $SU(N_f) \to Sp(N_f/2)$. We will focus on (2.8) with the breaking to $O(N_f)$ because it exists for all even $N_f$ and it fits the discussion of $SO(N)$ in section 4 below.

Let us investigate the algebra formed by time-reversal $\mathcal{T}$ and the symmetry $\mathcal{C}_f$. Clearly on local operators constructed by polynomials in fields these operators commute. However, on monopole operators we find a more interesting result. Since $\mathcal{T}$ acts as (2.7), the $\mathcal{C}_f$ charge of a monopole operator (i.e. $\pm 1$) is changed by the action of $\mathcal{T}$. Thus $\mathcal{T}$ and $\mathcal{C}_f$ do not commute in a sector with monopole charge, but instead they obey the algebra

$$\mathcal{T}\mathcal{C}_f\mathcal{T}^{-1} = \mathcal{C}_f\mathcal{M} \,. \tag{2.9}$$

One implication of this symmetry algebra is that

$$(\mathcal{C}_f\mathcal{T})^2 = \mathcal{T}^2\mathcal{M} \,. \tag{2.10}$$

The operator $\mathcal{C}_f\mathcal{T}$ is another antiunitary symmetry that reverses the orientation of time and hence gives another time-reversal symmetry of this theory. What we see from (2.9) is that one or the other of $\mathcal{T}^2$ or $(\mathcal{C}_f\mathcal{T})^2$ must always involve the magnetic symmetry $\mathcal{M}$.

## 2.1 $N_f = 2$: $O(4)$ Unitary Symmetry

The simplest model that exhibits $\mathcal{T}^2 = \mathcal{M}$ is the case $N_f = 2$. This model is special because it has been conjectured to flow to an infrared fixed point with unitary $O(4)$ symmetry [13,21, 29,30]. Here we will discuss the interplay between the enhanced symmetry and time-reversal.

The basis for the claim that the theory has enhanced global symmetry in the IR is a conjectural self-duality [13,29], which acts on the $(SU(2) \times U(1)_M)/\mathbb{Z}_2$ global symmetry discussed in the previous section. Specifically, the duality exchanges a $U(1)$ subgroup of the $SU(2)$ symmetry that acts on the fundamental fermions with the $U(1)_M$ magnetic symmetry. Since the former is part of an $SU(2)$ the latter must be as well.

More precisely, the duality in question states the equivalence of long-distance limits of the following two Lagrangians [13][2]

$$i\overline{\psi}^1 \slashed{D}_{a+X}\psi_1 + i\overline{\psi}^2 \slashed{D}_{a-X}\psi_2 - \frac{1}{4\pi}ada + \frac{1}{2\pi}adY + \frac{1}{4\pi}YdY$$

$$\longleftrightarrow \quad i\overline{\chi}^1 \slashed{D}_{\widetilde{a}+Y}\chi_1 + i\overline{\chi}^2 \slashed{D}_{\widetilde{a}-Y}\chi_2 - \frac{1}{4\pi}\widetilde{a}d\widetilde{a} + \frac{1}{2\pi}\widetilde{a}dX + \frac{1}{4\pi}XdX . \qquad (2.11)$$

In the above $a, \widetilde{a}$ are dynamical $U(1)$ gauge fields,[3] and $\psi, \chi$ are Dirac fermions. The fields $X$ and $Y$ are background $U(1)$ gauge fields coupling to the global symmetries that are exchanged under the duality. In the first line $Y$ couples to the magnetic symmetry and $X$ couples to the charged fields, while in the dual Lagrangian on the second line their roles are reversed.

In order to determine the enhanced IR symmetry and the properties of the time-reversal symmetry $\mathcal{T}$ in this model, we must first describe the complete UV symmetries. We use the language of the first line in (2.11). In addition to the $(SU(2) \times U(1)_M)/\mathbb{Z}_2$ symmetry described in general in the previous section, there is also an order two charge conjugation symmetry $\mathcal{C}$ that acts as

$$\mathcal{C}(\psi) = \overline{\psi} , \qquad \mathcal{C}(a) = -a , \qquad \mathcal{C}(X) = -X , \qquad \mathcal{C}(Y) = -Y . \qquad (2.12)$$

The theory also has time-reversal symmetry $\mathcal{T}$ with $\mathcal{T}(a) = a, \mathcal{T}(Y) = -Y$. We also define $\epsilon^X$ be the order four element in $SU(2)$ given by the matrix $\epsilon^X = \begin{pmatrix} 0 & 1 \\ -1 & 0 \end{pmatrix}$. These discrete symmetries of the UV theory are summarized in table 2.

| Symmetry | $a$ | $X$ | $Y$ | $\widetilde{a}$ |
|:---:|:---:|:---:|:---:|:---:|
| $\mathcal{C}$ | $-1$ | $-1$ | $-1$ | $-1$ |
| $\epsilon^X$ | $+1$ | $-1$ | $+1$ | $-1$ |
| $\mathcal{C}^Y \equiv \mathcal{C}\epsilon^X$ | $-1$ | $+1$ | $-1$ | $+1$ |
| $\mathcal{T}$ | $+1$ | $+1$ | $-1$ | $-1$ |

Table 2: Symmetries and their eigenvalues in $U(1)_0$ with two fermions of charge one. Note that $Y$ is charged under $\mathcal{T}$, and that $\mathcal{C}^Y\mathcal{T}$ commutes with the unitary global symmetry. Under the duality (2.11), $X \longleftrightarrow Y$ and $a \longleftrightarrow \widetilde{a}$.

Consider in particular the element $\mathcal{C}^Y$ defined above. This stabilizes the $SU(2)$, but acts on the $U(1)_M$ magnetic symmetry as reflection. Moreover, using the quotient described in (2.2), we see that $(\mathcal{C}^Y)^2 = (-1)^M$. It follows that including charge conjugation extends $U(1)_M$ to the group $Pin^-(2)_M$. Thus the unitary global symmetry in the ultraviolet is [30][4]

$$\frac{SU(2) \times Pin^-(2)}{\mathbb{Z}_2} . \qquad (2.13)$$

---

[2]Below and in the following we omit gravitational Chern-Simons terms in our description of dualities.
[3]More precisely they are $spin^c$ connections [25].
[4]This corrects a small misidentification of the global symmetry group in [21].

Now let us consider the implication of the self-duality (2.11). Since the duality exchanges the Cartan subgroup of the $SU(2)$ with the $U(1)_M$ magnetic symmetry, it is clear that (2.13) must be enhanced in the IR to a group where the two factors in the numerator are on equal footing. The simplest possibility is $O(4)$

$$\text{UV}: \frac{SU(2) \times Pin^-(2)}{\mathbb{Z}_2} \longrightarrow \text{IR}: O(4) . \tag{2.14}$$

Note that the exchange of the two $SU(2)$ subgroups is now implemented by the duality which acts as a global symmetry.

The group $O(4)$ has a $\mathbb{Z}_2$ center subgroup with non-trivial element $z$. From the point of view of the first duality frame in (2.11) we recognize that $z = (-1)^M$. Using our analysis of monopole operators in the previous section we therefore have:

$$\mathcal{T}^2 = (\mathcal{C}\mathcal{T})^2 = z . \tag{2.15}$$

This is consistent with the duality, which acts on the discrete symmetries in table 2 as

$$\mathcal{C} \longleftrightarrow \mathcal{C} , \qquad \mathcal{T} \longleftrightarrow \mathcal{C}\mathcal{T} , \qquad \epsilon^X \longleftrightarrow \mathcal{C}\epsilon^Y , \qquad \mathcal{C}\epsilon^X \longleftrightarrow \epsilon^Y . \tag{2.16}$$

# 3   QED$_3$ with Fermions of Even Charge

In this section we consider quantum electrodynamics with a single fermion $\psi$ of general even charge $q$. Since the charge is even, we can adjust the bare Chern-Simons level to achieve a time-reversal invariant theory $U(1)_0$. These theories have unitary symmetry $U(1)_M \rtimes \mathbb{Z}_2^{\mathcal{C}}$.

As in our analysis in section 2, our goal is to elucidate the properties of $\mathcal{T}$. On the local operators built from polynomials in the fields we find as usual $\mathcal{T}^2 = (-1)^F$. (In fact all the elementary gauge invariant local operators are bosons.) Thus we now turn to the monopole operators. The analysis is similar to that of the previous section, and hence we will be brief. For a complete treatment see [2].

In the background of a monopole of unit charge, the field $\psi$ now has $q$ zero modes, which form an irreducible representation under the Lorentz group of spin $j = (q-1)/2$. Notice that since $q$ is even, the modes carry half-integral spin. These zero modes act on the bare monopole state, $|0\rangle$, which has zero spin and electric charge $k_{bare} = -q^2/2$. The gauge invariant monopole operator $\mathfrak{M}$ is dressed by $q/2$ zero modes and hence we deduce that its statistics is correlated with the electric charge as

$$\mathfrak{M} \text{ is} \begin{cases} \text{fermionic} & q = 2 \bmod 4 , \\ \text{bosonic} & q = 0 \bmod 4 . \end{cases} \tag{3.1}$$

Meanwhile, we can also compute the sign produced by $\mathcal{T}^2$ acting on monopole operators. In this case, the modes fill out a Dirac spinor of $Spin(2q)$, and $\mathcal{T}^2$ on these states is $+1$ if the spinor is real, and $-1$ if the spinor is pseudoreal. We can compare this to the statistics of the monopole and we find, for all charge $q$, the expected relation $\mathcal{T}^2 = (-1)^F$ . Thus the algebra of symmetries involving time-reversal is simple

$$\mathcal{T}^2 = (-1)^F , \qquad \mathcal{T}\mathcal{C}\mathcal{T}^{-1} = \mathcal{C} , \qquad \mathcal{T}\exp(i\alpha M)\mathcal{T}^{-1} = \exp(-i\alpha M) . \tag{3.2}$$

Notice also that for $q = 2 \bmod 4$ the only fermionic operators are those with odd monopole number and hence $(-1)^F = (-1)^M$, while for $q = 0 \bmod 4$ all gauge invariant local operators are bosons.

Let us also discuss the possible anomalies of the time-reversal symmetry $\mathcal{T}$. We can compute the time-reversal anomaly $\nu$ valued in $\mathbb{Z}_{16}$ by counting Majorana fermions $\lambda$ in the UV lagrangian. In this calculation, a given fermion can have a sign $\sigma$ that appears in the formula $\mathcal{T}(\lambda) = \sigma \gamma_0 \lambda$, and the contribution to $\nu$ of $\lambda$ is $\sigma$. This leads to the formulas

$$\nu_{\mathcal{T}} = 2 \,, \qquad \nu_{\mathcal{CT}} = 0 \,. \tag{3.3}$$

### 3.1 Infrared Behavior

The models discussed above have simple long-distance description that can be derived assuming the particle-vortex dualities studied in [5,31–33]. This duality states that the following two Lagrangians describe the same IR physics (note the carefully normalized coefficients [33]):

$$i\overline{\psi}\slashed{D}_A\psi - \frac{1}{4\pi}AdA \quad \longleftrightarrow \quad i\overline{\chi}\slashed{D}_a\chi - \frac{1}{2\pi}adb + \frac{2}{4\pi}bdb - \frac{1}{2\pi}bdA \,. \tag{3.4}$$

In the above, our conventions are such that lower case letters (such as $a$, $b$) indicate dynamical abelian gauge fields, while capital letters (such as $A$) indicate classical background fields that couple to global symmetry currents.

Let us briefly summarize several aspects of this duality. The right-hand-side above is an interacting Chern-Simons matter theory with $\chi$ a Dirac fermion of charge one. In particular it defines a non-trivial RG flow. Meanwhile, the left-hand-side is free theory of a Dirac fermion $\psi$, which can therefore be viewed as the long-distance limit of the interacting theory. Under the duality, the magnetic global symmetry of the interacting theory is exchanged with the flavor symmetry in the dual free Dirac description. Thus, the operator $\psi$ is dual to the monopole operator in the interacting description.

#### 3.1.1 Duality for a Charge Two Fermion

We now assume (3.4) and use it to derive the IR behavior of $U(1)_0$ coupled to an even charge fermion beginning with the case $q = 2$.

We substitute $A \to 2U$ and add classical terms $\frac{2}{4\pi}UdU + \frac{1}{2\pi}UdB$ on both sides of (3.4), where $U$ and $B$ are new background fields. We then set $U \to u$ and promote $u$ to be dynamical. On the right-hand-side, if we change variables to $\widehat{u} = u - b$ then the field $b$ becomes a Lagrange multiplier that can be integrated out. This results in the fermion-fermion duality[5]

$$i\overline{\psi}\slashed{D}_{2u}\psi - \frac{2}{4\pi}udu + \frac{1}{2\pi}udB \quad \longleftrightarrow \quad i\overline{\chi}\slashed{D}_B\chi + \frac{2}{4\pi}\widehat{u}d\widehat{u} + \frac{1}{2\pi}\widehat{u}dB \,. \tag{3.5}$$

The left-hand side above is QED with charge-two Dirac fermion $\psi$. In this duality frame, the classical field $B$ couples to the $U(1)_M$ magnetic global symmetry. The dual description on the right-hand side is a free Dirac fermion and the TQFT $U(1)_2$, which we can view as the IR limit of the interacting theory. In this frame $B$ couples to the $\chi$ flavor symmetry and to the $U(1)_2$ sector. Thus, as in the particle-vortex duality (3.4), the monopole operator becomes a free field at long distances.

Notice that both sides of the duality have $\mathcal{T}$ symmetry. On the left-hand side this is simply because the bare Chern-Simons level for the dynamical gauge field $u$ has been adjusted to make the theory time-reversal invariant. On the right-hand side the time-reversal symmetry exists because level-rank duality $U(1)_2 \longleftrightarrow U(1)_{-2}$ (see table 1). Although both theories are time-reversal invariant, their two antiunitary symmetries are exchanged under the duality

$$\mathcal{T} \quad \longleftrightarrow \quad \mathcal{CT} \,. \tag{3.6}$$

---

[5]We can repeat this analysis with the substitution $A \to A + 2U$ and then make $U$ dynamical, while keeping a nontrivial background $spin^c$ connection $A$. Then we have a duality similar to (3.5), which is valid on a $spin^c$ manifold. However, if we do that the time-reversal symmetry is modified to $\mathcal{T}(A) = A$, $\mathcal{T}(B) = -B + 2A$.

This can be seen by comparing the transformation properties of the various gauge fields. For instance on the left-hand side $\mathcal{T}(u) = u$ and hence $\mathcal{T}(B) = -B$. Therefore the dual description of this symmetry must act with a minus sign on $\widehat{u}$ and hence is $\mathcal{CT}$.

We can also compare the time-reversal anomalies for these theories across the duality. In the description as $U(1)_0$ with a charge two fermion the anomalies are read off from the Lagrangian resulting in (3.3). This matches with the free Dirac description if we use the fact that the anomaly for the semion-fermion spin TQFT $U(1)_2$ is

$$\nu_{\mathcal{T}}(U(1)_2) = -2 , \qquad \nu_{\mathcal{CT}}(U(1)_2) = 2 , \tag{3.7}$$

where $\mathcal{T}$ and $\mathcal{CT}$ denote two distinct ways that the time-reversal anyon permutation symmetry can couple to the theory (they are also called $SF_-$ and $SF_+$ in the literature) [4, 8, 16].

One way to check the duality (3.5) is to deform both sides by relevant operators. Assuming that the RG flow is smooth, the resulting theories after deformation must still be dual. Across the duality (3.5) the fermion mass terms map to each other with a relative sign $\overline{\psi}\psi \longleftrightarrow -\overline{\chi}\chi$. For positive coefficient of $\overline{\psi}\psi$, the two sides of the duality flow to $U(1)_2$ coupled to a background magnetic gauge field. For negative coefficient of $\overline{\psi}\psi$ the duality becomes:

$$-\frac{2}{4\pi}udu + \frac{1}{2\pi}udB \quad \longleftrightarrow \quad \frac{2}{4\pi}\widehat{u}d\widehat{u} + \frac{1}{2\pi}\widehat{u}dB + \frac{1}{4\pi}BdB . \tag{3.8}$$

Again, these two theories are equivalent via level-rank duality.

It is useful to explore how this theory and its long-distance behavior are modified when we add monopole operators to the Lagrangian. These results will also enable us to anticipate many features of the higher rank $SO(N)+$ tensor gauge theories described in section 5.

We perturb the theory by a bosonic operator of monopole charge two [20]:[6] $\delta\mathcal{L} = i\mathfrak{M}^2 + h.c.$. This interaction breaks the $U(1)_M$ magnetic symmetry down to $\mathbb{Z}_2$ generated by $(-1)^M \equiv \mathcal{M}$. This interaction also breaks the symmetry $\mathcal{T}$. However it preserves a new symmetry

$$\mathcal{T}' \equiv \mathcal{T}e^{i\pi M/2} . \tag{3.9}$$

It is straightforward to determine the algebra of the unbroken symmetries $\mathcal{C}, \mathcal{M}, \mathcal{T}'$ using their embedding in the algebra (3.2). We have

$$\mathcal{T}'\mathcal{C}\mathcal{T}'^{-1} = \mathcal{T}e^{i\pi M/2}\mathcal{C}e^{-i\pi M/2}\mathcal{T}^{-1} = (\mathcal{T}e^{i\pi M/2}\mathcal{T}^{-1})(\mathcal{T}\mathcal{C}\mathcal{T}^{-1})(\mathcal{T}e^{-i\pi M/2}\mathcal{T}^{-1}) = \mathcal{CM} . \tag{3.10}$$

Thus the algebra of symmetries is now non-abelian. By similar manipulations we can also determine that

$$(\mathcal{T}')^2 = (-1)^F , \qquad (\mathcal{CT}')^2 = (-1)^F\mathcal{M} . \tag{3.11}$$

Note also that the time-reversal anomaly $\nu_{\mathcal{T}'}$ is simply equal to $\nu_{\mathcal{T}}$ since the operators only differ by a magnetic symmetry in the UV. Meanwhile for the antiunitary symmetry $\mathcal{CT}'$ the anomaly $\nu$ is no longer meaningful.

We can also find the same result using the infrared description (3.5). The UV monopole operator interaction maps to a mass term $(\chi_1)^2 - (\chi_2)^2$, where we have written the complex fermion in terms of Majorana components. This mass term breaks the flavor symmetry down to a $\mathbb{Z}_2$ subgroup. Each mass term $\chi_i^2$ is odd under $\mathcal{T}$, and hence time-reversal is also broken. However, the combination of $\mathcal{T}$ with a flavor rotation by $\pi/2$ is preserved and is identified with $\mathcal{T}'$.

The effect of this mass term is to split the Dirac point into two distinct Majorana points. The phase in the middle is $U(1)_2 \cong SO(2)_2$ as illustrated in figure 1.

---

[6]In general we can add the operator $\alpha\mathfrak{M}^2 + h.c.$ for complex coefficient $\alpha$. Here we consider the $\mathcal{C}$-even monopole perturbation. If instead we add a $\mathcal{C}$-odd deformation, then we find equivalent physics. Specifically, the perturbed theory preserves $\mathcal{C}' = \mathcal{C}e^{\pi i M/2}$ and $\mathcal{T}$, and the two symmetries again do not commute: $\mathcal{T}^{-1}\mathcal{C}'\mathcal{T} = \mathcal{C}'\mathcal{M}$.

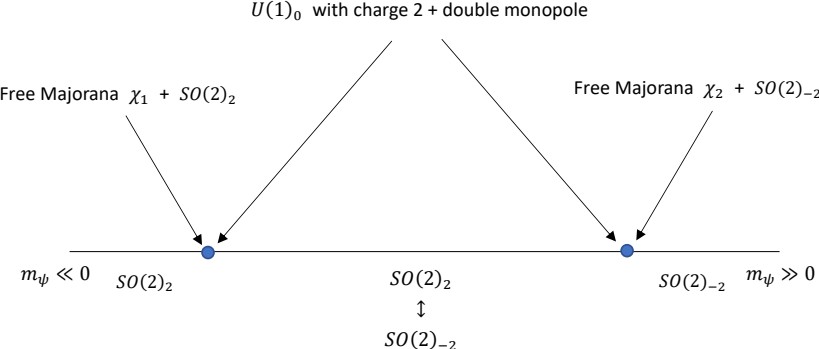

Figure 1: The phase diagram of $U(1)$ gauge theory coupled to a Dirac fermion with charge two, together with a double monopole perturbation. The monopole perturbation splits the free Dirac point into two Majorana points. These transitions separate TQFTs.

### 3.1.2 Duality for General Even Charge

We now extend our analysis to theories with a general even charge $q$. Starting from (3.5) we add the classical terms $-\frac{q/2}{2\pi}BdX + \frac{1}{2\pi}Xd\widehat{B}$ to both sides. We then set $B \to b$ and $X \to x$ with $x, b$ dynamical (on the right we also rename $x$ as $\widehat{x}$). On the left-hand side, the integral over $b$ is trivial leading to the duality

$$i\overline{\psi}\slashed{D}_{qx}\psi - \frac{q^2/2}{4\pi}xdx + \frac{1}{2\pi}xd\widehat{B} \longleftrightarrow i\overline{\chi}\slashed{D}_b\chi + \frac{2}{4\pi}\widehat{u}d\widehat{u} + \frac{1}{2\pi}\widehat{u}db - \frac{q/2}{2\pi}bd\widehat{x} + \frac{1}{2\pi}\widehat{x}d\widehat{B} \, . \tag{3.12}$$

Note that in the special case $q = 2$, we can also integrate out $\widehat{x}$ on the right and reproduce the duality (3.5).

The left-hand side of the duality (3.12) is QED with even charge $q$ fermion $\psi$. The right-hand side is a free Dirac fermion $\chi$ together with a $U(1)_2$. The field $\widehat{x}$ is a Lagrange multiplier, which reduces $b$ to a $\mathbb{Z}_{q/2}$ gauge field which couples to both $\chi$ and the topological sector $U(1)_2$. On local operators the effect of the $\mathbb{Z}_{q/2}$ gauge field is simply to quotient the spectrum. In particular, the right-hand side is effectively free and hence can be viewed as the IR limit of the interacting theory on the left-hand side.

Many of the essential features of this duality are similar to the case of charge two.

- The unit charge monopole operator $\mathfrak{M}$ in the QED description maps across the duality to the operator $\chi^{q/2}$, which is the minimal local operator consistent with the $\mathbb{Z}_{q/2}$ quotient. Note (via integrating out $\widehat{x}$) that both operators couple to the background field $\widehat{B}$ with unit charge and that the statistics of these operators agree from our general result (3.1).

- Both theories are time-reversal invariant. Across the duality $\mathcal{T}$ and $\mathcal{CT}$ are exchanged and the time-reversal anomalies agree using (3.7).

- The fermion mass terms match again up to a relative sign. Deforming the duality by these relevant operators we find that both theories flow to the TQFT $U(1)_{\pm q^2/2}$.

As a further consistency check of the general charge $q$ duality, we can match the one-form symmetry and its 't Hooft anomaly. Both theories in (3.12) have $\mathbb{Z}_q$ one-form symmetry [35]: on the left it is generated by $x \to x + \frac{1}{q}d\theta$ for periodic scalar $\theta \sim \theta + 2\pi$, while on the right

it is generated by $\widehat{x} \to \widehat{x} + \frac{1}{q}d\theta$ and $\widehat{u} \to \widehat{u} + \frac{1}{2}d\theta$. We can turn on background gauge field $B_2$ for this one-form symmetry and study its 't Hooft anomaly. Since the fermion mass term is invariant under the one-form symmetry, the anomaly can be computed from the resulting TQFT $U(1)_{\pm q^2/2}$ under the mass perturbation $m\overline{\psi}\psi$ with $m$ positive or negative. This gives the same $\mathbb{Z}_2 \subset \mathbb{Z}_q$ valued anomaly on both sides of the duality

$$\pi \int_Y \frac{\mathcal{P}(B_2)}{2} \, , \tag{3.13}$$

where $Y$ is a closed four-manifold, $\mathcal{P}$ is the Pontryagin square with coefficient in $\mathbb{Z}_q$, and $B_2$ is the background two-form $\mathbb{Z}_q$ gauge field for the one-form symmetry.

### 3.1.3 Monopole Deformation of the Charge Four Theory

Let us now specialize from general charge $q$ and consider some aspects of the theory with $q = 4$. This is the same theory as $Spin(2)_0$ coupled to a symmetric tensor fermion [15] and hence our results here anticipate the higher-rank generalizations of section 5.

Again we find it useful to add a monopole operator interaction to break the magnetic $U(1)_M$ symmetry. In the $Spin(2)_0$ theory the unit charge monopole $\mathfrak{M}$ of the $SO(2)_0$ theory is absent and instead, the basic allowed monopole operator is the charge two monopole $\mathfrak{M}^2$ of $SO(2)_0$. As in the analysis of section 3.1.1, we add the perturbation $\delta\mathcal{L} = i\mathfrak{M}^2 + h.c.$, but in this case, the $U(1)_M$ symmetry is completely broken.

This deformation breaks the time-reversal symmetry $\mathcal{T}$, but preserves the antiunitary symmetry $\mathcal{T}' = \mathcal{T}e^{i\pi M}$. Therefore after deformation the unbroken symmetries are $\mathcal{T}'$ and charge conjugation $\mathcal{C}$. By using the algebra (3.2) we find that after the deformation the symmetry algebra is standard

$$\mathcal{T}'\mathcal{C}\mathcal{T}'^{-1} = \mathcal{C} \, , \qquad \mathcal{C}^2 = 1 \, , \qquad \mathcal{T}'^2 = (-1)^F \tag{3.14}$$

Moreover, the time-reversal anomalies are unmodified from their values before the symmetry breaking perturbation, i.e. $\nu_{\mathcal{T}'} = \nu_{\mathcal{T}}$ and $\nu_{\mathcal{C}\mathcal{T}'} = \nu_{\mathcal{C}\mathcal{T}}$.

The long-distance behavior for the theory with $q = 4$ and the symmetry breaking monopole perturbation can be obtained by gauging the $\mathbb{Z}_2$ magnetic symmetry $\mathcal{M}$ in the phase diagram of the theory with $q = 2$ presented in figure 1. In the IR, the monopole deformation is a mass term $(\chi_1)^2 - (\chi_2)^2$ for the two Majorana fermions, and integrating out these massive fields generates a non-trivial Lagrangian for the new $\mathbb{Z}_2$ gauge theory. Specifically this is the theory $(\mathbb{Z}_2)_1$, where the subscript indicates that this is the minimal consistent level in $\mathbb{Z}_2$ gauge theory. (See [15] for additional discussion.)

Taking into account the new topological sector from gauging $\mathcal{M}$ we find that in the presence of the monopole perturbation, the infrared theory is the TQFT

$$\frac{U(1)_8 \times (\mathbb{Z}_2)_1}{\mathbb{Z}_2} \quad \longleftrightarrow \quad O(2)_{2,1} \, , \tag{3.15}$$

where the quotient on the left-hand side gauges the one-form symmetry generated by the product of a charge 4 line in $U(1)_8$ and the Wilson line of the $\mathbb{Z}_2$ gauge theory. This is equivalent to the T-Pfaffian spin TQFT [22, 23], and it is also dual to $O(2)_{2,1}$ [15].

In particular, the time-reversal anomalies $\nu_{\mathcal{C}\mathcal{T}} = 0$, $\nu_{\mathcal{T}} = 2$ agree with those of T-Pfaffian$_+$ (the subscript indicates a particular definition of the antiunitary symmetry). Note that the names $\mathcal{T}, \mathcal{C}\mathcal{T}$ are reversed in the literature, see [4, 16] for $\nu_{\mathcal{C}\mathcal{T}}$ for T-Pfaffian, and [3, 5] for $\nu_{\mathcal{T}}$. The latter symmetry of T-Pfaffian is the diagonal subgroup of the conventional time-reversal symmetry and the magnetic symmetry of $O(2)$. (In the duality $U(1)_8 \longleftrightarrow O(2)_2$ the charge conjugation symmetry of $U(1)_8$ maps to the magnetic symmetry of $O(2)_2$ [15].)

The resulting phase diagram is illustrated in figure 2

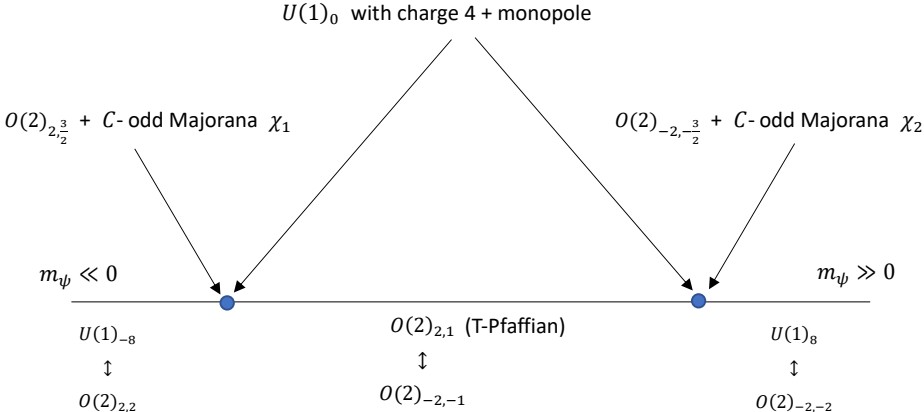

Figure 2: The phase diagram of $U(1)$ gauge theory coupled to a Dirac fermion of charge four, with a minimally charged magnetic monopole perturbation. In the two dual descriptions the Majorana fermion $\chi$ couples to the $\mathbb{Z}_2$ gauge field of $O(2)$ by the transformation $\chi \to -\chi$. The low energy TQFT is the T-Pfaffian theory.

# 4   $SO(N)_0$ with Vector Fermions

In this section we consider $SO(N)_0$ with $N_f$ fermions in the vector representation, where $N_f$ is even to avoid the standard parity anomaly. These theories have a time-reversal symmetry $\mathcal{T}$, whose basic properties we discuss below. In section 2 we analyzed the case $N = 2$, and many features of those models are common to the case $N > 2$.

We first describe the global unitary symmetries. There is a flavor symmetry $O(N_f)$, which includes a flavor charge conjugation $\mathcal{C}_f$ that acts on one of the flavor indices as reflection. There is also charge conjugation $\mathcal{C}$ and a $\mathbb{Z}_2$ magnetic symmetry $\mathcal{M}$.[7] Notice that in comparison with the abelian gauge theories analyzed in section 2 the magnetic symmetry in $SO(N)$ for $N > 2$ is always $\mathbb{Z}_2$ (measured by the $\mathbb{Z}_2$-valued second Stiefel-Whitney class). Thus the global symmetries agree with those of $U(1)_0$ with $N_f$ flavors after the deformation by a monopole operator of charge two.

Now let us consider the time-reversal symmetry $\mathcal{T}$. Clearly on local operators constructed out of the elementary fields we find the standard algebra (i.e. $\mathcal{T}^2 = (-1)^F$). Meanwhile on monopole operators the analysis of time-reversal symmetry and its square is similar to that of section 2. The basic monopole can be described by a gauge field configuration with unit flux in $SO(2) \subset SO(N)$, and each of the $N_f$ fermions in the vector representation has one zero mode.

Therefore, we similarly find that the time-reversal symmetry $\mathcal{T}$ does not commute with $\mathcal{C}_f$ on operators carrying magnetic charge

$$\mathcal{T} \mathcal{C}_f \mathcal{T}^{-1} = \mathcal{C}_f \mathcal{M} . \tag{4.1}$$

In addition we have

$$\mathcal{T}^2 = \begin{cases} (-1)^F \mathcal{M} & N_f = 2 \bmod 4 , \\ (-1)^F & N_f = 0 \bmod 4 , \end{cases} \qquad (\mathcal{C}_f \mathcal{T})^2 = \begin{cases} (-1)^F & N_f = 2 \bmod 4 , \\ (-1)^F \mathcal{M} & N_f = 0 \bmod 4 . \end{cases} \tag{4.2}$$

---

[7]Depending on $N_f$ and $N$, there can be discrete identifications on these global symmetries when acting on gauge invariant local operators.

We can also compute the time-reversal anomaly $\nu$ of these theories. For an antiunitary symmetry that squares to $(-1)^F$, $\nu$ is given by the number of Majorana fermions $\psi$ that transform as $\psi(x,t) \to \gamma^0 \psi(x,-t)$ minus the number of Majorana fermions $\psi'$ that transform as $\psi'(x,t) \to -\gamma^0 \psi'(x,-t)$. Therefore

$$\nu_{\mathcal{T}} = NN_f \;, \qquad \nu_{\mathcal{C}_f \mathcal{T}} = NN_f - 2N \;. \tag{4.3}$$

Notice that it is not obvious that the answer (4.3) is gauge invariant since the sign in the time-reversal transformation of a fermion can be changed by a gauge symmetry. Consider for instance combining $\mathcal{T}$ or $\mathcal{C}_f \mathcal{T}$ with the $\mathbb{Z}_2 \subset SO(N)$ gauge transformation $\text{diag}(1, \cdots -1, \cdots)$ with $2p$ total minus signs. The time-reversal anomalies (4.3) change to

$$\Delta \nu_{\mathcal{T}} = 4N_f p \;, \qquad \Delta \nu_{\mathcal{C}_f \mathcal{T}} = 4(N_f - 2)p \;. \tag{4.4}$$

However as emphasized above, the anomaly $\nu$ of an antiunitary symmetry is only meaningful when that symmetry squares to fermion parity. According to (4.2) when this is so, the ambiguity above vanishes in $\mathbb{Z}_{16}$, and the anomaly $\nu$ is well-defined.

As a final remark on these models, let us consider gauging the magnetic symmetry $\mathcal{M}$. This changes the gauge group from $SO(N)$ to $Spin(N)$ [15]. From (4.2), we conclude that in the $Spin(N)_0$ gauge theory coupled to vector fermions both $\mathcal{T}$ and $\mathcal{C}_f \mathcal{T}$ square to $(-1)^F$. In particular, the anomaly $\nu$ is meaningful for both symmetries.[8] This is also compatible with the calculation (4.4). The group $Spin(N)$ is a double covering of $SO(N)$, and the gauge transformation used in (4.4) is an $\mathbb{Z}_2$ element only if $p$ is even.

# 5 $SO(N)_0$ with Two-Index Symmetric Tensor Fermion

For our final class of models, we consider $SO(N)_0$ with one fermion in the two-index symmetric tensor representation. Across the transition where the fermion becomes massless the Chern-Simons level jumps from $-(N+2)/2$ to $+(N+2)/2$. Therefore, $N$ must be even to achieve a time-reversal invariant theory at zero fermion mass. Aspects of these theories were discussed in [15, 20]. The special case $N = 2$ is $U(1)$ gauge theory coupled to a charge two fermion and was analyzed in section 3.

These models have unitary global symmetry $\mathcal{C}$ and $\mathcal{M}$ which form $\mathbb{Z}_2 \times \mathbb{Z}_2$, as well as time-reversal symmetry $\mathcal{T}$. Notice that these are the symmetries present in $U(1)$ plus a charge two fermion, after deformation by a monopole operator of magnetic charge two. Therefore many of aspects of these models are similar.

## 5.1 Time-Reversal Symmetry and its Anomaly

As in all our previous analysis, on elementary fields the time-reversal symmetry $\mathcal{T}$ satisfies $\mathcal{T}^2 = (-1)^F$ and hence we turn to the sector with non-trivial monopole charge. The bare classical monopole operator transforms in the $(N+2)/2$-index symmetric tensor representation of $SO(N)$. This can be derived, for instance, by giving the fermions mass and using

$$k_{bare} = -\frac{N+2}{2} \;. \tag{5.1}$$

---

[8]More generally, whenever we have the algebra $\mathcal{T}^2 = (-1)^F X$ with some $X$ we could try to gauge the symmetry generated by $X$ to find a new theory with $\mathcal{T}^2 = (-1)^F$. However, there is a subtlety in doing it. A mixed anomaly between $\mathcal{T}$ and $X$ in the original theory can mean that after gauging $\mathcal{T}$ is no longer a symmetry. More precisely, as we said in the introduction, the gauging of $X$ leads to a one-form global symmetry generated by $\widehat{X}$ and the mixed anomaly can lead to a new symmetry, a 2-group, which mixes $\mathcal{T}$ and $\widehat{X}$ [36]. We will not analyze it in detail here. We simply point out that in the example above and in section 5.3 this phenomenon does not happen.

If $N = 0 \bmod 4$, this representation is charged under the center of $SO(N)$ and cannot be screened by any elementary fermion field. Thus in this case, there is no local monopole operator. By contrast when $N = 2 \bmod 4$, the bare monopole is neutral under the center of $SO(N)$ and hence can be dressed by fermions to form a gauge invariant local operator.

Regardless of whether the charge of the monopole can be screened by fundamental fields, we can always produce sectors with monopole charge by attaching an appropriate Wilson line to classical configuration. The resulting object is gauge invariant, but nonlocal since it now contains the line.

The algebra of global symmetries in sectors with magnetic charge can again be understood by analyzing zero modes in a monopole background. To be specific, we consider a unit magnetic flux in the $N, N-1$ direction of the gauge group which breaks $SO(N)$ to $(O(2) \times O(N-2))/\mathbb{Z}_2$. The symmetric tensor fermion decomposes in this background into the following fields:

- A Dirac fermion of charge 2 under $O(2)$ which is neutral under $O(N-2)$. This field has two zero modes which form a spin 1/2 doublet. We indicate them by $\psi_a$ for $a = 1, 2$. There are also complex conjugate fields.

- A Dirac fermion of charge 1 under the $O(2)$ which transforms as a vector of $O(N-2)$. This field has $N-2$ zero modes which are Lorentz scalars. We denote them via their embedding in the symmetric tensor as $(\psi^{(N,N-j)} + i\psi^{(N-1,N-j)})$, where $j = 1, \cdots N-2$. There are also complex conjugate fields.

- $(N^2 - 3N + 2)/2$ fermions which are neutral under the $O(2)$. These fields have no zero modes in the monopole background and hence decouple from our analysis.

Let $|0\rangle$ indicate the bare monopole described above. Quantizing the zero-modes leads to a Hilbert space of states

$$|0\rangle , \quad \cdots , \quad \psi_1 \psi_2 \prod_{j=2}^{N-1} (\psi^{(N,N-j)} + i\psi^{(N-1,N-j)})|0\rangle , \tag{5.2}$$

any of which can be made gauge invariant by attaching a suitable Wilson line. The action of time-reversal symmetry $\mathcal{T}$ exchanges the top and bottom states listed in (5.2). Notice that charge conjugation $\mathcal{C}$ acts by a sign on fields with gauge index one. Therefore, the bottom and top states above have opposite charge under $\mathcal{C}$, and we see that the algebra of symmetries is non-abelian

$$\mathcal{T}\mathcal{C}\mathcal{T}^{-1} = \mathcal{C}\mathcal{M} . \tag{5.3}$$

The non-abelian algebra above is closely related to the behavior of discrete $\theta$-parameters discussed in [15]. Specifically, for any level $k$ we can consider the $SO(N)_k$+tensor fermion theory coupled to a background $\mathbb{Z}_2$ gauge field $B^{\mathcal{C}}$ for the charge conjugation global symmetry. As the mass of the tensor is transitioned from negative to positive the effective action shifts by the coupling $\pi \int_X B^{\mathcal{C}} \cup w_2$ where $w_2$ is the second Stiefel-Whitney class of the $SO(N)$ bundle, which measures the charge $\mathcal{M}$. This means that across such a tensor transition the symmetry $\mathcal{C}$ is exchanged with $\mathcal{C}\mathcal{M}$. In the specific case of a time-reversal invariant theory this implies (5.3) since the fermion mass is odd under $\mathcal{T}$.

We can also compute the action of $\mathcal{T}^2$ and $(\mathcal{C}\mathcal{T})^2$. Since the monopole is effectively abelian $\mathcal{T}^2$ is fixed by the parity of the bare Chern-Simons level as described in section 1.3. Using the formula (5.1) we conclude that

$$\mathcal{T}^2 = \begin{cases} (-1)^F & N = 2 \bmod 4 , \\ (-1)^F \mathcal{M} & N = 0 \bmod 4 , \end{cases} \qquad (\mathcal{C}\mathcal{T})^2 = \begin{cases} (-1)^F \mathcal{M} & N = 2 \bmod 4 , \\ (-1)^F & N = 0 \bmod 4 . \end{cases} \tag{5.4}$$

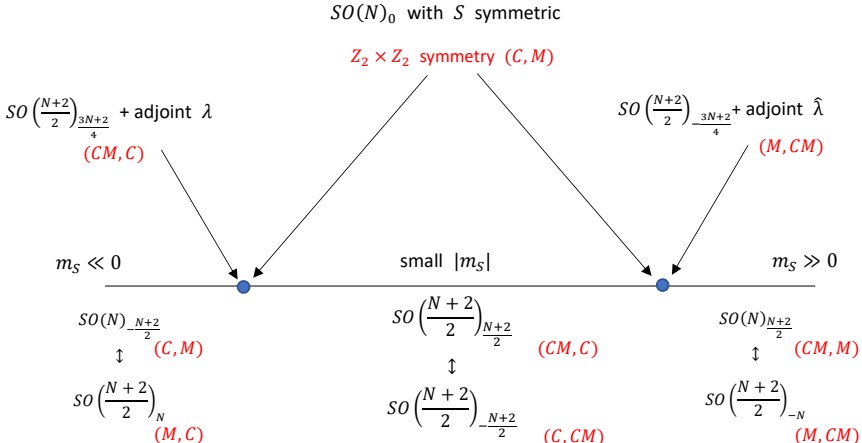

Figure 3: The phase diagram of $SO(N)$ gauge theory coupled a symmetric tensor fermion $S$. The infrared TQFTs, together with relevant level-rank duals are shown along the bottom. The blue dots indicate the transitions from the semiclassical phase to the quantum phase. Each of these transitions can be described by a dual theory with adjoint fermions, in which the transition can be seen at weak coupling. These dual theories cover part of the phase diagram. These figures are identical to those in [20], with the map of the $\mathbb{Z}_2 \times \mathbb{Z}_2$ unitary global symmetry determined from [15].

It is also straightforward to compute the time-reversal anomaly $\nu$ for these theories [20]

$$\nu_{\mathcal{T}} = \frac{N^2 + N - 2}{2} , \qquad \nu_{\mathcal{CT}} = \frac{N^2 - 3N + 2}{2} . \tag{5.5}$$

As a consistency check, consider mixing $\mathcal{T}$ or $\mathcal{CT}$ with a $\mathbb{Z}_2 \subset SO(N)$ gauge transformation $\mathrm{diag}(1, \cdots -1, \cdots)$ with $2p$ minus signs. The change in the anomalies is

$$\Delta \nu_{\mathcal{T}} = 8p^2 - 4Np , \qquad \Delta \nu_{\mathcal{CT}} = 8p^2 + 8p - 4Np . \tag{5.6}$$

Exactly when $\mathcal{T}$ or $\mathcal{CT}$ square to fermion parity, these changes above vanish modulo sixteen as expected.

## 5.2 Time-Reversal Symmetry in the IR

The long-distance behavior of these models has been analyzed in [20] leading to the proposed phase structure summarized in figure 3.

In particular, for small fermion mass the theory is conjectured to flow to a quantum phase described by the spin TQFT $SO(n)_n$ with $n = (N+2)/2$. This theory is time-reversal invariant as a spin TQFT by level-rank duality [14] (see table 1)

$$SO(n)_n \quad \longleftrightarrow \quad SO(n)_{-n} . \tag{5.7}$$

Let $\mathcal{T}^{\mathrm{IR}}$ be the time-reversal symmetry of the infrared TQFT that squares to $(-1)^F$. It is related to the UV symmetries discussed in the previous section via

$$\mathcal{T}^{\mathrm{IR}} = \begin{cases} \mathcal{T}^{\mathrm{UV}} & N = 2 \bmod 4 , \\ \mathcal{C}^{\mathrm{UV}} \mathcal{T}^{\mathrm{UV}} & N = 0 \bmod 4 . \end{cases} \tag{5.8}$$

The time-reversal anomaly $\nu$ for the TQFT $SO(n)_n$ is known to be $n$ [19]. Combining this with the map (5.8) of UV and IR symmetries we can check the phase diagram of figure 3 using anomaly matching. We have:

$$N = 2 \bmod 4 \implies \nu_{\text{UV}} - \nu_{\text{IR}} = \nu_{\mathcal{T}^{\text{UV}}} - n = \frac{N^2 + N - 2}{2} - \frac{N+2}{2} = \frac{N^2 - 4}{2}, \tag{5.9}$$

$$N = 0 \bmod 4 \implies \nu_{\text{UV}} - \nu_{\text{IR}} = \nu_{\mathcal{C}^{\text{UV}}\mathcal{T}^{\text{UV}}} - n = \frac{N^2 - 3N + 2}{2} - \frac{N+2}{2} = \frac{N^2 - 4N}{2}.$$

Both expressions vanish modulo sixteen as expected.

The charge conjugation and magnetic symmetries in the UV and IR are related by

$$\mathcal{M}^{\text{UV}} = \mathcal{C}^{\text{IR}}\mathcal{M}^{\text{IR}}, \quad \mathcal{C}^{\text{UV}} = \mathcal{C}^{\text{IR}}. \tag{5.10}$$

In particular, the algebra (5.3) for $\mathcal{C}^{\text{UV}}, \mathcal{M}^{\text{UV}}$ and $T^{\text{UV}}$ implies in the infrared the time-reversal symmetry satisfies

$$\mathcal{T}^{\text{IR}}\mathcal{C}^{\text{IR}} = \mathcal{M}^{\text{IR}}\mathcal{T}^{\text{IR}}, \tag{5.11}$$

which is consistent with the fact that $\mathcal{C}^{\text{IR}}, \mathcal{M}^{\text{IR}}$ are exchanged under level-rank duality [14,15].

## 5.3 Gauging the Magnetic Symmetry: $Spin(N)_0$ + Tensor Fermion

Consider gauging the $\mathbb{Z}_2$ magnetic symmetry $\mathcal{M}$. This changes the theory to $Spin(N)_0$ coupled to a symmetric tensor fermion. These models are a natural generalization of the $U(1)_0$ with a charge four fermion discussed in section 3.1.3. From (5.4) we immediately see that both time-reversal symmetries $\mathcal{T}$ and $\mathcal{C}\mathcal{T}$ are standard (see footnote 8)

$$\mathcal{T}\mathcal{C}\mathcal{T}^{-1} = \mathcal{C}, \quad \mathcal{T}^2 = (\mathcal{C}\mathcal{T})^2 = (-1)^F. \tag{5.12}$$

In particular, the anomaly $\nu$ for both $\mathcal{T}$ and $\mathcal{C}\mathcal{T}$ is well-defined for all even $N$. This is compatible with the anomaly computation (5.6) since there $p$ is required to be even for the $\mathbb{Z}_2$ gauge transformation to be an element of $Spin(N)$.

The phase diagram of the $Spin(N)$ gauge theory was derived in [15] and is reproduced in figure 4. This is related to the phase digram in figure 3 by gauging. In particular, for small fermion mass the theory is conjectured to flow to a quantum phase described by the spin TQFT $O(n)^1_{n,n-1}$ (the notation as in [15]) with $n = (N+2)/2$, which is time-reversal invariant as a spin TQFT by level-rank duality [15] (see table 1)

$$O(n)^1_{n,n-1} \quad \longleftrightarrow \quad O(n)^1_{-n,-n+1}. \tag{5.13}$$

This theory is a higher rank generalization of the T-Pfaffian theory $O(2)_{2,1}$ discussed in section 3.1.3 for $N = 2$. (See also appendix H of [15].)

From the phase diagram in figure 3 we can deduce that the unitary symmetries in the UV and IR are related as

$$\mathcal{C}^{\text{UV}}(Spin(N)) = \mathcal{M}^{\text{IR}}(O(n)). \tag{5.14}$$

Thus from (5.8), the antiunitary symmetries in the UV and IR are related as

$$\mathcal{T}^{\text{IR}} = \begin{cases} \mathcal{T}^{\text{UV}} & N = 2 \bmod 4, \\ \mathcal{C}^{\text{UV}}\mathcal{T}^{\text{UV}} & N = 0 \bmod 4, \end{cases} \qquad \mathcal{M}^{\text{IR}}\mathcal{T}^{\text{IR}} = \begin{cases} \mathcal{C}^{\text{UV}}\mathcal{T}^{\text{UV}} & N = 2 \bmod 4, \\ \mathcal{T}^{\text{UV}} & N = 0 \bmod 4. \end{cases} \tag{5.15}$$

We proceed to match the anomaly $\nu$ between the short-distance and long-distance descriptions. The anomaly for $\mathcal{T}^{\text{IR}}$ always matches the corresponding UV anomaly, since they agreed

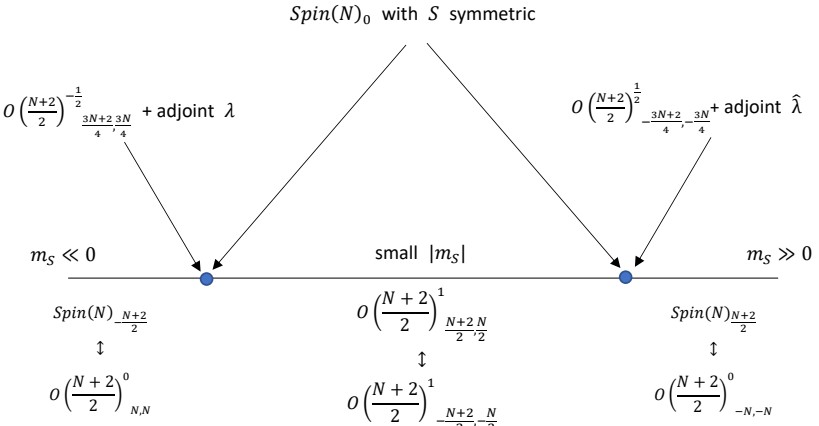

Figure 4: The phase diagram of a $Spin(N)$ gauge theory coupled to a fermion $S$ in the two-index symmetric tensor representation. It can be obtained from $SO(N)$ gauge theory by gauging the magnetic symmetry in the UV, which corresponds to gauging the diagonal $\mathcal{CM}$ in the long-distance TQFT. The blue dots indicate the transitions from the semiclassical phase to the quantum phase. Each of these transitions can be described by a dual theory with adjoint fermions, in which the transition can be seen at weak coupling. These dual theories cover part of the phase diagram. These figures are identical to those in [15].

in the $SO(N)$ theory before gauging $\mathcal{M}^{\text{UV}}$ [20]. Therefore, we focus on the anomaly for the antiunitary symmetry $\mathcal{M}^{\text{IR}}\mathcal{T}^{\text{IR}}$.

The UV computation is as in the $SO(N)$ theories (5.5). In the IR we need to compute $\nu_{\mathcal{MT}}(O(n)^1_{n,n-1})$. This can be done using the formalism in [8, 17]. The answer depends on choices we do not know how to determine, like the eigenvalue of $\mathcal{T}^2$ on some anyons and on a choice of orientation (i.e. the sign of $\nu$).

We will split the discussion depending on $N \bmod 4$. For $N = 0 \bmod 4$, $n$ is odd and we have (see appendix D of [15]):

$$O(n)^1_{n,n-1} \quad \longleftrightarrow \quad SO(n)_n \times (\mathbb{Z}_2)_{2(n-1)} \,. \tag{5.16}$$

Furthermore, for odd $n$ the magnetic symmetry in $O(n)^1_{n,n-1}$ does not permute the anyons [15], and thus both $\mathcal{T}^{\text{IR}}$ and $\mathcal{M}^{\text{IR}}\mathcal{T}^{\text{IR}}$ define the same permutation action on the anyons. Therefore, the anomaly $\nu$ can be obtained from that of $SO(n)_n$ [19] and that of $(\mathbb{Z}_2)_0, (\mathbb{Z}_2)_4$ [3, 4, 8]

$$\nu((\mathbb{Z}_2)_0) = 0 \text{ or } 8 \,, \qquad \nu((\mathbb{Z}_2)_4) = 0 \text{ or } +4 \text{ or } -4 \,, \tag{5.17}$$

where the values depend on the choices mentioned above. With appropriate choices here we match that anomalies with those of the UV theory!

For $N = 2 \bmod 4$, $n$ is even and the magnetic symmetry of $O(n)^1_{n,n-1}$ permutes the anyons. This makes the computation of the anomaly slightly more involved and we have not carried it out explicitly. Instead, we use the anomaly matching with the UV theory to conjecture that for even $n$ we have $\nu_{\mathcal{MT}}(O(n)^1_{n,n-1}) = \pm 5(n-2)$. Note that for $n = 2$ this agrees with the expected answer for the T-Pfaffian$_+$ theory. (This is to be contrasted with the known value $\nu_{\mathcal{T}} = \pm n$ of these theories.)

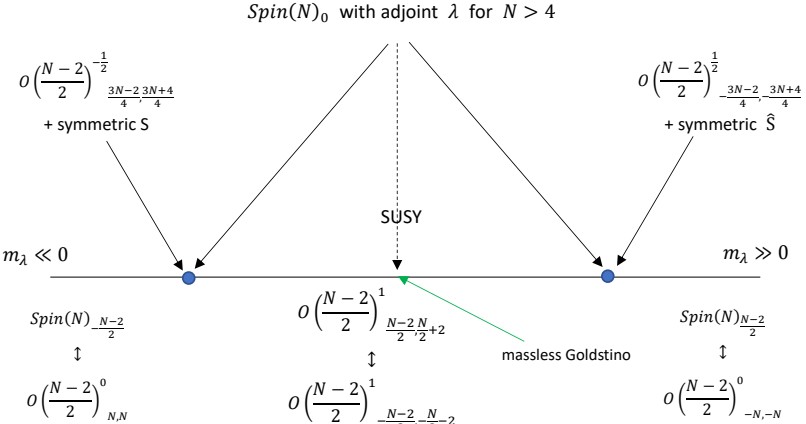

Figure 5: The phase diagram of a $Spin(N)$ gauge theory coupled to a fermion $\lambda$ in the adjoint representation. The phase transitions are visible at weak coupling in a dual theory with symmetric tensor fermions. This can be derived from the $SO(N)$ gauge theory by gauging the UV magnetic symmetry $\mathcal{M}^{\mathrm{UV}}$, which corresponds to gauging the symmetry $\mathcal{C}^{\mathrm{IR}}\mathcal{M}^{\mathrm{IR}}$ in the long-distance TQFT [15].

The entire discussion of this section can be repeated with a fermion in the adjoint representation. In particular, for gauge group $SO(N)$ these theories also have the non-commutative algebra of symmetries (5.3). Similar phase diagrams for $SO(N)$ and $Spin(N)$ gauge groups are conjectured in [15, 20]. Now the infrared theory consists of a massless Goldstino (take $N > 4$) and a time-reversal invariant TQFT. For gauge group $Spin(N)$ the TQFT is $O(n)^1_{n,n+3}$ with $n = (N-2)/2$ (see figure 5), and we can match both $\nu_{\mathcal{T}}$ and $\nu_{\mathcal{CT}}$ between the UV and the IR. Again, the matching of $\nu_{\mathcal{T}^{\mathrm{IR}}}$ follows from the matching in $SO(N)$ [20]. The matching of $\nu_{\mathcal{M}^{\mathrm{IR}}\mathcal{T}^{\mathrm{IR}}}$ is a new test of the conjectured phase diagram. In particular, for $N = 0 \bmod 4$, $n$ is odd and we can use the relation $O(n)^1_{n,n+3} \longleftrightarrow SO(n)_n \times (\mathbb{Z}_2)_{2(n+1)}$ [15]. Then, with an appropriate choice in (5.17) we match the UV and the IR values.

# Acknowledgements

We thank M. Barkeshli, M. Cheng, J. Gomis, Z. Komargodski, M. Metlitski, X. Qi, T. Senthil, J. Wang, and E. Witten for discussions. C.C. is supported by the Marvin L. Goldberger Membership at the Institute for Advanced Study, and DOE grant DE-SC0009988. The work of P.-S.H. is supported by the Department of Physics at Princeton University. The work of N.S. is supported in part by DOE grant DE-SC0009988.

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
