# Peer review of "Time-Reversal Symmetry, Anomalies, and Dualities in (2+1)$d$"

_SciPost Physics, doi:SciPost Phys. 5, 006 (2018)_

## Round 2 · Referee Report · Anonymous · 2018-5-31

Strengths

-1- The questions investigated in the paper are subtle, and are treated with great care and detail.

-2- The paper is very well written, difficult issues are explained with great clarity.

Weaknesses

-1- Just before eq. 2.13, the authors write "It follows that including charge conjugation extends U(1)M to the group Pin−(2)M". It would be nice to expand a bit the discussion, and explain in more detail the similarities/differences with previous analysis of the UV and IR global symmetry of QED with 2 flavors, of refs [30] and [21].

Report

The paper considers various examples of Quantum Field Theories invariant under time-reversal, such that the time-reversal operator satisfies an interesting non-standard algebra.

Requested changes

-1- It looks like the variable M appearing in the r.h.s. of eq 1.10 is not defined in the introduction.

-2- I found a few typos:
- last line of sec 1.1: "their"
- 5th to last line of sec 1.2: "instance"
- eq 3.2: curly C vs standard C
- sec 3.1.1, 2nd line: "beginning"

---

## Round 2 · Referee Report · Anonymous · 2018-6-2

Strengths

1. Addresses subtle details of time-reversal symmetry in three-dimensional gauge theories
2. Timely subject matter
3. Clarity of presentation

Weaknesses

None, really.

Report

The point of this paper is to work out various aspects of time-reversal symmetry and the exact global symmetry group for particular $T$-invariant Chern-Simons-matter theories in three dimensions. The basic question is exactly what bosonic symmetry does $T$ square to. In some examples (like 3d QED coupled to bosons) it squares to 1, and in others with charged fermions it squares to $(-1)^F$. Most interestingly, there are examples demonstrated by the authors where $T^2$ involves the monopole number symmetry.

More generally the authors work out in some detail the global symmetry group of these Chern-Simons-matter theories, see e.g. (1.15). The way they do it is straightforward enough to follow and reproduce. The non-trivial part is to deduce how the global symmetries act on monopole operators, and for this one simply looks at matter zero-mode creation/annihilation operators in the monopole background as around (2.3). The presentation is a little sparse on the details, but there is more than enough information given to work it out on one's own.

Beyond logical completeness, there is some payoff from these largely formal manipulations. The two I noticed were:

1. In examples where $T^2$ involves the monopole symmetry, one must now be careful about the computation of the time-reversal anomaly $\nu$. There may be mixed anomalies between $T$ and $(-1)^M$.

2. There has been a conjectured self-duality between 3d QED with 2 fermions, whereby the global symmetry is enhanced to $O(4)$ in the infrared. The authors find additional evidence for this conjecture by carefully tracking down how $T$ acts.

Requested changes

I have two extremely pedantic suggestions. The first is that the authors should explain in-text the notation for O(n) with a superscript and two subscripts in Table 1. The second is that in various places the authors refer to the "global symmetry algebra" when they mean the group.

---

## Editorial Decision

published